# A Project Based Learning (PBL) Approach Involving PET Recycling in Chemical Engineering Education

**Krishna Prasad Rajan** [1] , **Aravinthan Gopanna** [2] **and Selvin P. Thomas** [1,2,*]

[1] Department of Chemical Engineering Technology, Yanbu Industrial College, Royal Commission Yanbu Colleges and Institutes, PO Box-30436, Yanbu 41912, Saudi Arabia; rajank@rcyci.edu.sa

[2] Advanced Materials Laboratory, Yanbu Research Center, Royal Commission Yanbu-Colleges and Institutes, PO Box 30436, Yanbu 41912, Saudi Arabia; gopannaa@rcyci.edu.sa

\* Correspondence: thomass@rcyci.edu.sa; Tel.: +966-551960211

**Abstract:** The recycling of waste plastics is considered as one of the strategies to tackle the issue of environmental pollution caused by commodity plastics all over the world. Recently, many universities have incorporated topics related to recycling and plastics waste management into their curricula at different levels to increase awareness as well as to develop new recycling technologies. In this study, one of the most important waste recycling problems is given as the project for the undergraduate students of chemical engineering to analyze the effectiveness of the project-based learning (PBL) approach in the school curriculum. A team of students was assigned with the task of recycling post-consumer polyethylene terephthalate (PET) bottles through an experimental and design approach. From the experimental data, students designed a recycling plant with a proposed capacity to produce 1 ton of recycled granules per day through the project-based learning approach. Evaluation of the project was carried out at various stages and it was found that the students acquired the required skills and applied them effectively. The outcomes of the present study clearly establish that the problems which have societal impacts, such as waste management, environmental pollution, etc., can be effectively communicated to the student community through the PBL approach, which can lead to increased motivation and enhanced critical thinking abilities.

**Keywords:** project-based learning; chemical education research; recycling; scale-up; evaluation strategies

## 1. Introduction

Modern engineering education is designed to deliver knowledge and skills to young learners, such as theoretical backgrounds in mathematics, physics, and chemistry, along with professional practices of engineering tools and methods. Mere theoretical knowledge of the curriculum would not make a successful engineer, and due to the rapid advancements in the field of science and technology, even a fresh graduate is expected to show readiness to accept challenging tasks in a professional manner. In recent years, the engineering profession and the bodies responsible for accrediting engineering programs are advocating for more effective strategies to impart the technical knowledge and relevant skills to the student community undergoing engineering education. In response to the call for a change from the conventional "chalk and talk strategy" of teaching and learning, experts in the field of professional technical education developed various strategies. Many learning methods are proposed to develop the additional skills in graduating engineers and the project-based learning approach is one of the most sought after pedagogical methods these days.

Learning through projects is considered as one of the pedagogical strategies in student-centered learning in which the students are exposed to real-world skills such as problem-solving, critical thinking, experimental designs, data collection, analysis and interpretation of the results, cooperative

working, and effective communication [1]. This learning strategy helps to engage a group of students and helps them to learn how to learn.

There are many learning styles, especially for engineering students. The conventional method of reading from textbooks and memorizing information will not help the students to achieve the required skill sets for an engineering career. Across the world, many new strategies were thought developed, and case studies and active learning were suggested as alternative learning strategies. The case study method was adopted basically in the social sciences and life sciences as it progresses through surveys among the stakeholders [2]. The conclusions are derived from the analysis of the various surveys and customers. However, it has a serious drawback, as the design is rigid, and the questions cannot be changed once the case is framed. Therefore, the actual results may not reflect the ground reality after the duration of the survey [3].

The active learning strategy involves students more directly in the learning process, as opposed to the conventional methods. English scholar R. W. Revans introduced the term active learning for the first time and described it as follows: "In active learning, students participate in the process, and students participate when they are doing something besides passively listening" [4].

There were many definitions of active learning. In short, active learning engages students by having them doing things related to learning and thinking about what they are doing while they are doing it [5]. Project-based learning is a well-known strategy for implementing active learning in engineering education in which a group of students completes a project. The project-based learning amalgamates knowing and doing. In this learning approach, students not only learn knowledge and skills mandated by the core curriculum but also apply what they know to solve a realistic problem which has relevance in the industry or society [6]. Generally, the project should contain a design, model, and simulation, etc. The completion of the project is by the submission of a written report followed by oral and poster presentations describing the major achievements of the project [7]. Project-based learning refocuses education on the student, not the curriculum, via the advisory role of the teaching staff rather than the complete authoritarian mode of teaching and learning, which is the shift mandated by the professional bodies [8].

In their criteria for accrediting engineering programs, many accreditation agencies for engineering education, including ABET, give importance to acquiring the required knowledge and skills, application of these skills to solve the problems in their core areas through tests, and analysis and interpretation [9]. These criteria also require the students to involve themselves in self-directed learning in their professional domain [10]. Through the project-based learning approach, along with the program courses, a graduate can assimilate the aforementioned criteria during his course of study of a particular program [11]. A student graduation project, normally offered during the last semester of the program, helps the student to apply the knowledge and skills that they have earned through various course components throughout the semesters of their study to a real-world project [12].

The project-based learning approach has been utilized for enhancing the skills of engineering students recently. Davies et al. at Massey University recently reported the use of the project-based learning method for different groups of final year engineering students on the topic of the preparation of biodiesel [13]. The students gained project management ability, communication, and presentation skills through the execution of the projects in the PBL method. Noordin and Nordin recently reported the usefulness of the project-based learning approach to improve the non-technical skills of engineering students in three Malaysian Universities through interviews and observations [14]. They have identified three main attributes—namely coaching and supervision, continuous assessment, and real-world experience that can improve the non-technical skills of an engineering student apart from the knowledge he gains from the technical aspects of the education. A recent report from San-Valero et al. also discussed the applicability of project-based learning approaches to improve non-technical skills for chemical engineering students [15]. Murat Genc reported on the effect of the project-based learning approach to improve attitudes toward the environment and environmental problems [16]. The students were well-equipped in identifying environmental problems and identifying apt solutions

for them through the PBL approach. The impact of the project-based learning approach on the thinking of undergraduate students, their ability to solve problems, and successful implementation of the suggested solutions, etc., were reported by Chamberlain and Mendoza recently [17]. Another example of the successful implementation of the PBL approach is the reported development of electrical vehicles by students of the electrical engineering department [18].

This paper discusses how the project-based learning approach is applied in chemical engineering education through a case study of a project given to the final semester students of a baccalaureate degree program in chemical engineering at a technical institution in Saudi Arabia. The project theme was to design a plant for recycling post-consumer polyethylene terephthalate (PET) bottles to PET granules with a proposed daily capacity of 1 ton.

## 2. Project Description

Recycling of plastics waste is considered an important technique for addressing the serious concerns associated with the environmental pollution arising out of dumping waste plastics into the municipal solid waste (MSW) stream. The growing awareness of the public about the seriousness of this issue and strict government regulations also helped to explore various recycling techniques to handle waste plastics [19]. As per a recent report, almost 5.25 trillion plastic particles weighing 268,940 tons are currently floating in the sea [20]. It is also predicted that on the continuation of this trend, by 2050, the sea will contain more plastic waste than fish [21]. The major chunk of waste plastic that accumulates in the ecosystem is comprised of plastic bottles made of polyethylene terephthalate (PET). Thus, the recycling of PET bottles is an imperative process both from an economic as well as an environmental point of view. Like any other polymers, the recycling of PET can also be divided into four categories. These include primary recycling, mechanical recycling or secondary recycling, chemical recycling or tertiary recycling, and finally, energy recovery or quaternary recycling [22]. More details about the various recycling techniques can be found elsewhere [23–25].

In mechanical recycling, the waste polymer (PET), which is usually contaminated with food wastes, adhesives, and labels, are cleaned by a thorough washing and drying process. The shredded PET wastes are then melt-processed using a conventional extrusion technique to produce recycled granules. In the extrusion, plastic material in the form of small flakes is gravity fed from a top mounted hopper into the barrel of the extruder. The molten polymer exits the extruder die in the form of strands which then pass through a water bath. After proper drying, the strands are then cut into the required dimensions after passing through a pelletizer. A typical flow diagram depicting the entire process is shown in Figure 1.

The extrusion process is the most important step in the mechanical recycling of waste PET bottles to produce recycled granules. Therefore, calculations pertaining to the design and operation of the extruder play a major role in the recycling process. Performing experimental trials in a laboratory scale extruder and scaling-up the process to a higher capacity extruder is an attractive approach for the design of a bigger system, depending on the requirements of the production of the recycled granules. Carley et al. [26] predicted the performance of large-scale extruders from the performance data obtained from geometrically similar extruders with smaller capacities. A more refined and recent study on the scale-up of extruder parameters to define the geometry and operation conditions of a target extruder, provided the material is being subjected to similar flow and heat transfer conditions, was carried out by Covas and Cunha [27]. In the present project, the recycling of waste PET bottles was carried out in a laboratory scale extruder, and from the obtained data, calculations were performed to scale-up the process to get one ton of recycled granules per day. The duration of the project was 15 instructional weeks. The group was provided with all the necessary library facilities, including access to all major electronic databases, to understand the recycling processes in detail and to come up with a suitable project plan. The group was allowed to use all the facilities of a polymer processing laboratory.

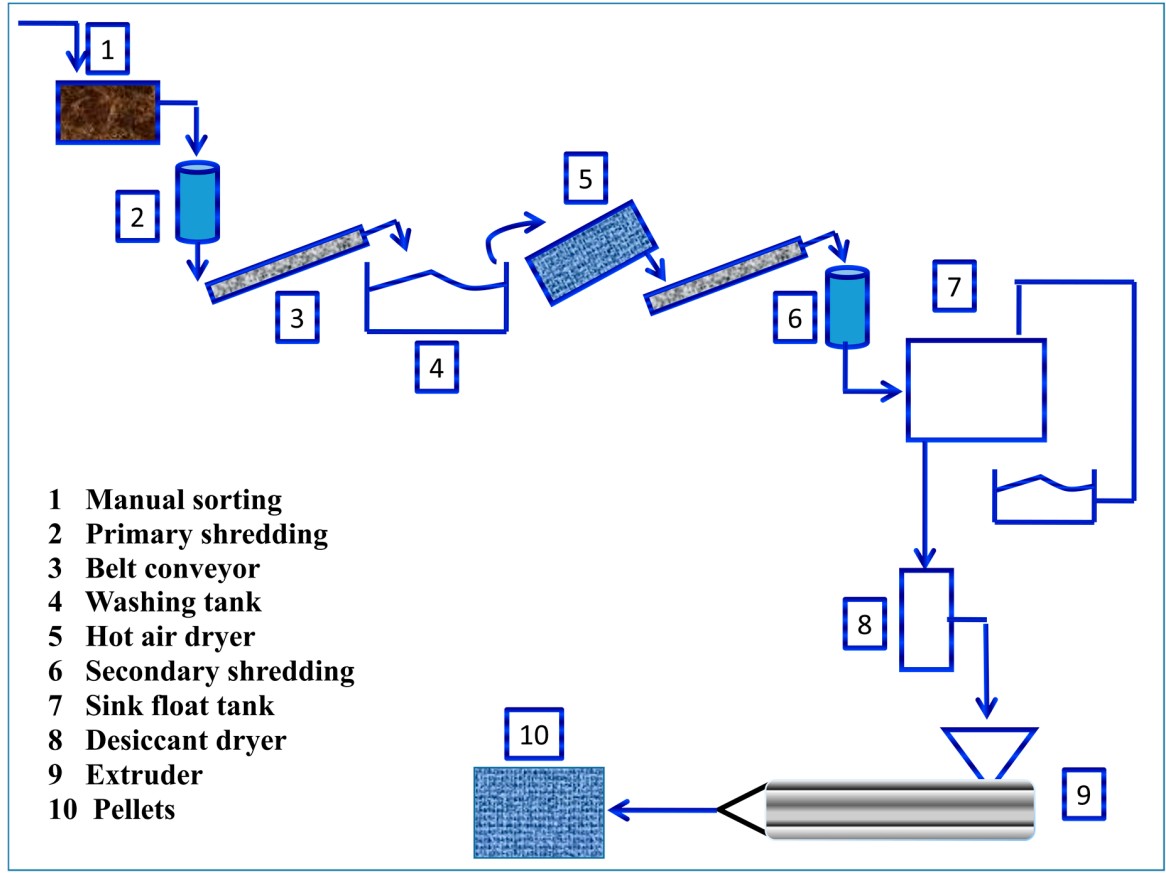

**Figure 1.** Typical flow diagram for a polyethylene terephthalate (PET) recycling process.

*Methodology*

The students were directed to conduct an extensive literature review on polymer extrusion, polymer materials, recycling, plant design, scale-up calculations, and technical report writing. Based on the gathered knowledge and ideas, the group was instructed to submit a preliminary project design. As part of the design, equipment, process variables, and experimental details were reviewed by the team and laboratory experiments were performed. Process optimization was carried out based on the results obtained from the previous stages. An interim report with all the obtained data was submitted to the department through the supervisor. Scale up and design calculations were performed to obtain the required output of one ton per day of recycled granules. Feedback and necessary modifications were provided by the supervisor at various stages starting from the design conceptualization to the stage of scale-up calculations. The team was solicited to provide the final recommendation based on the results and calculations. Finally, a comprehensive technical report was prepared as per the stipulated guidelines and submitted to the evaluation committee. The evaluation was carried out according to the criteria set forth by the department. A simplified flow diagram on the methodology of the project is shown in Figure 2.

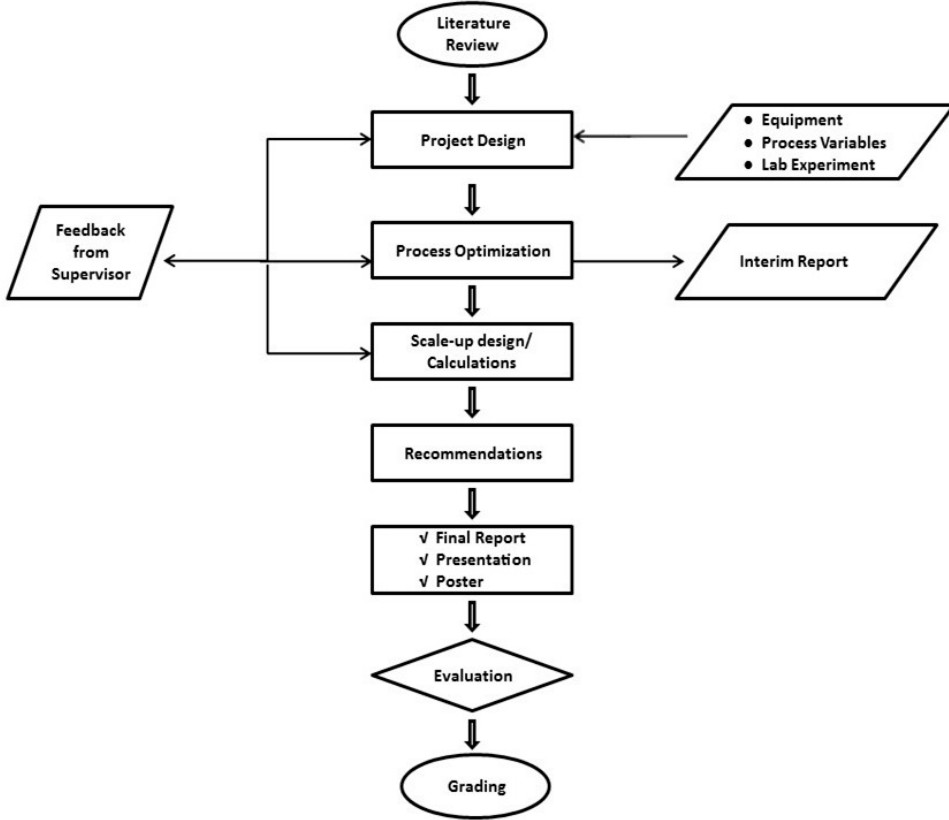

**Figure 2.** Diagram showing the overall methodology of the process execution.

## 3. Materials and Methods

Used PET water bottles (1-liter capacity, colorless) were collected and cleaned thoroughly after removing the labels and neck ring. The bottles selected were of Aquafina 1-liter bottled drinking water to obtain uniformity in the raw material characteristics. The bottles were dried in sunlight for 8 h to completely remove the water. The dried bottles were cut into small flakes of 5 mm size using a bottle scrap cutting machine (custom made, chamber size 230 × 200 mm, with 2 stationary and 9 rotary cutters and power 10 kW). The flakes were dried in a hot air oven at 120 °C for 6 h before the extrusion. The details of the extruder used in the study are given in Table 1.

**Table 1.** Extruder specifications.

| Machine Model | Haake PolyLab OS Rheomex 19/33 |
|---|---|
| Machine type | Modular torque rheometer system equipped with single screw extruder. |
| Screw diameter | 19.05 mm |
| L/D ratio | 33 |
| Maximum screw speed | 250 rpm |
| Maximum temperature | 450 °C |
| Maximum pressure | 700 bar |
| Maximum torque | 160 Nm |
| Mains supply | 3 × 400 V |

### 3.1. Extruder Control and Process

There are four temperature zones in the extruder. These zones are referred as Z1, Z2, Z3, and Z4. The temperature settings applied to various temperature zones during the experiment are given in Table 2.

After pre-drying at 120 °C for six hours, the PET flakes were fed to the hopper of the extruder. The molten polymer passing through the die exit was directed to a water bath to cool it down. After drying

properly, the strands were pelletized. A set of experiments were conducted with varying screw speeds to understand the optimum speed, according to the output obtained from these experiments. From the obtained data, scale-up calculations were performed to design an extruder required to produce 1 ton of recycled PET granules per day.

**Table 2.** Profile settings.

| Extruder Temperature Zone | Set Temperature (°C) |
|:---:|:---:|
| Z1 | 120 |
| Z2 | 240 |
| Z3 | 260 |
| Z4 | 270 |

*3.2. The Evaluation Strategy*

The criteria and weighting for the assessment of the project were defined and informed to the students in the first week, as shown in Table 3.

**Table 3.** For assessment of the project.

| Sl. No. | Task | Weightage (%) |
|:---:|:---:|:---:|
| 1. | Identifying the project task and defining the underlying problem | 10 |
| 2. | Literature review | 10 |
| 3. | Experimental design and implementation | 20 |
| 4. | Analysis and interpretation of the results | 20 |
| 5. | Plant layout | 10 |
| 6. | Preparation of reports | 15 |
| 7. | Peer review | 5 |
| 8. | Project presentation | 10 |

## 4. Project Results

The experiments were conducted continuously for three hours before the representative experimental data were collected. The outputs obtained from these experiments are given in Table 4.

**Table 4.** Experimental data.

| Speed (N, rpm) | Time (min) | Output Q (g/min) |
|:---:|:---:|:---:|
| 200 | 1 | 38 |
| 225 | 1 | 42 |
| 250 | 1 | 48 |

*Scaling-Up*

The scaling up of the extrusion process from the data obtained from a small-scale laboratory extruder is a challenging task. From the machine parameters and the experimental results, the screw design calculations required to scale up the process to obtain the desired capacity (1 ton/day) were performed [28,29]. A screw speed of 250 rpm was considered for the calculations, as the output was maximum at that speed. The approach used is given in Table 5.

The scale-up factors for the extruder used to produce the required output (1 ton/day) was calculated according to Table 4. The details of these calculations are given in Appendix A. The summary of the experiment parameters and the scale-up results are given in Table 6.

The recommendation suggested based on the experimental and calculated results is to utilize a commercially available extruder with a screw diameter 75 mm and L/D ratio 33:1. Such an extruder is capable of yielding the required output after 12 h of extrusion processing in a day. The granulation can be achieved either by adopting water cooling followed by pelletization, as employed in the laboratory studies or by using the die face cutter unit, depending on the customer demands.

**Table 5.** Scale-up factors for extrusion [28,29].

| Factor | Experimental Data | Scale-Up Factors |
|---|---|---|
| Diameter, D | $D_1$ | $D_2$ |
| Channel width, W | $W_1$ | $W_2 = W_1(D_2/D_1)$ |
| Channel depth, h | $h_1$ | $h_2 = h_1\sqrt{\left(\frac{D_2}{D_1}\right)}$ |
| Screw speed, N | $N_1$ | $N_2 = N_1\sqrt{\left(\frac{D_2}{D_1}\right)}$ |
| Volumetric output, Q | $Q_1$ | $Q_2 = Q_1\,(D_2/D_1)^2$ |
| Shear rate | $\gamma1$ | $\gamma2 = \gamma1\,(D_2/D_1)$ |
| Circumferential speed, V | $V_1$ | $V_2 = V_1\sqrt{\left(\frac{D_2}{D_1}\right)}$ |
| Residence time, $t_{res}$ | $t_1$ | $t_1\sqrt{\left(\frac{D_2}{D_1}\right)}$ |
| Screw power, Z | $Z_1$ | $Z_1(D_2/D_1)^{2.5}$ |

**Table 6.** Experiment data and scale up results.

| Parameters | Machine and Experiment Data | Scale up Result |
|---|---|---|
| D | 19.05 mm | 72.4 mm |
| W | 12.7 mm | 48.26 mm |
| H | 5.2 mm | 10.14 mm |
| N | 250 rpm | 487.3 rpm |
| Q | 48 g/min | 694 g/min |
| Γ | 2876 $min^{-1}$ | 5606 $min^{-1}$ |
| V | 14954 mm/min | 29150 mm/min |
| $t_{res}$ | 20.6 min | 40.16 min |
| Z | 2.4 kW | 67.56 kW |

## 5. Discussion

Senior design projects are offered during the final semester of the undergraduate program with a pre-defined set of learning outcomes. The outcomes are formulated keeping in mind the required knowledge and skill sets pertaining to the core area of specialization, as well as the cognitive and psychomotor skills outlined in the respective program outcomes. Typical learning outcomes for a design project in chemical engineering are as follows:

a. Define and analyze a contemporary chemical engineering problem and apply knowledge and skills to solve it.
b. Design and conduct experiments and analyze and interpret data.
c. Design a system, component, or process to meet the desired needs within realistic constraints such as economic, environmental, social, political, ethical, health and safety, manufacturability, and sustainability constraints.
d. Utilize issues and problems from different disciplines such as other engineering fields, businesses, etc.
e. Organize records on project advancements.
f. Observe professional and ethical responsibilities and safety rules and regulations.
g. Discuss the impact of the engineering solutions in global, economic, environmental, and social contexts.
h. Recognize the need to engage in life-long learning and develop the ability to do so.
i. Discuss contemporary issues in a chosen area.
j. Use the techniques, skills, and modern engineering tools necessary for engineering practices.
k. Communicate effectively.

The project is evaluated at various stages by the supervisor and the committee formed by experts in the field. The supervisor evaluates the project continuously by communicating with the participants and monitoring their progress. The committee is formed by inviting one external expert, usually from the nearby industry, and the other from academia. The industry expert is chosen based on his experience in handling real-time projects in the workplace. The expert from the academia is chosen based on his expertise in the respective project. This expert can be from the same department or from other departments of the college or also from another university/institute. The committee evaluates the project based on three aspects—technical report, oral presentation, and poster presentation. The technical report is evaluated based on the criteria put forward at the beginning of the project itself and an average of the points given by the examiners are considered for finalizing the grades. The draft report of the project is to be submitted to the examiners by week 14 of the semester and the committee will evaluate and submit the comments during the oral presentation in week 15. Based on the comments and suggestions, the group has to submit the final report to the institution during week 16. The oral presentation will be done for a stipulated time period, and each participant will be asked to present a portion of the work. Managing the total time allotted, clarity of the work done, delivery of the data and results, body language, and finally, the question and answers related to the project, ethics, and teamwork are evaluated individually. A similar strategy is also applied in the case of poster presentation.

The project selected is an ideal one in order to assess the listed outcomes for a typical undergraduate design project. The assessment of the diverse criteria was done according to a grading scale from 1–5. The different grades were distributed as excellent (5), above average (4), average (3), below average (2), and poor (1). The description of each grade according to the selected tasks is given in Table 7. The assessments of the different tasks are done by the supervisor and this constitutes 70% of the total grade.

**Table 7.** Criteria for each grade.

| Task | Excellent | Above Average | Average | Below Average | Poor |
|------|-----------|---------------|---------|---------------|------|
| Identifying the project task and defining the underlying problem | The scope of the project work is detailed clearly. The goals and motivation of the project are well defined. A well-crafted problem statement and planning of the project with the Gantt chart is provided. A feasibility study of the product or scale-up is proposed or performed. | The scope of the project is detailed. The goals and motivation of the project are well defined. A well-crafted problem statement and planning of the project with the Gantt chart is provided. A feasibility study of the scale-up or the product is not provided. | The scope of the project is given briefly. The goals and motivation of the project are not defined. Problem statement and planning are vague. No statement about scale-up of product development. | The scope of the project is not clear. The goals and motivation of the project are not defined. Problem statement and planning are clumsy. No scale-up proposal or product development. | The scope of the project is missing. The goals and motivation are not defined. Problem statement and planning not provided. No scale up proposal or product development. |
| Literature review | Extensive research being done to find out the prior art. Resources such as the library, internet, text books, journals, magazines were explored extensively. A comprehensive collection of the literature and references is done. Self-motivated learning ability displayed. | A satisfactory research is done on prior art. Resources such as the library, internet, text books, journals, magazines were explored to an appropriate level. An adequate collection of the literature and references is done. Self-directed learning ability is shown in glimpses. | Prior art search is done to a minimum level. Only few resources were explored. Less than sufficient literature and references were collected. Requires motivating to collect the information. | Prior art search was done to a minimum level. Not much literature and references collected. Required strong motivation to find out the literature and references. | Absolutely no information or literature collected. No self-motivation to do the prior art search. Close to no collection of literature and references. |

**Table 7.** *Cont.*

| Task | Excellent | Above Average | Average | Below Average | Poor |
|---|---|---|---|---|---|
| Experimental design and implementation | Execute the most appropriate scientific principles in developing the experimental design. Plan the experimental works week by week, envisioning the results in each stages. Conduct the works as sub tasks and revisit at each stage for any upgradation. Innovation and creativity is shown in the implementation of the works. | Execute the most appropriate scientific principles in developing the experimental design. Plan the experimental works week by week. Conduct the works as sub tasks. Creativity is shown in the implementation of the works. | Execute the appropriate scientific principles in developing the experimental design but have hurdles in between. Plans the experimental works week by week but fails to accomplish them. Innovation and creativity are shown to a lesser extent in the implementation of the works. | Experimental design is developed with minimum scientific principles. Not much plan to do the experimental works. Not much Innovation and creativity is shown in the implementation of the works. | A causal approach to experimental design. No specific plan. No creativity is shown. |
| Analysis of the results | Experimental data were collected and analyzed using various means. Analyzed results were utilized for proposing the plant design and up-scaling. | Experimental data are collected and analyzed as appropriate. Few utilized for plant design and up-scaling. | Collected experimental data and minimum analyses done. Proposed the plant design without considering the actual results. | Obtained experimental data. Minimum analysis was done. | Experimental data could not be generated. |
| Plant layout | Establish and design the layout of a plant for the recycling of 1 ton of plastics per day.The use of advanced engineering tools, software, and simulation at competent levels without guidance is expected. | Establish and design the layout of a plant for recycling of 1 ton of plastics per day.The use of advanced engineering tools, software and simulation at a competent level with some guidance is expected. | Establish and design the layout of a plant for recycling of 1 ton of plastics per day.The use of advanced engineering tools, software and simulation in competent level with complete guidance is expected. | Designed the layout of a plant for recycling of 1 ton of plastics per day in an amateur style.The use of advanced engineering tools, software, and simulation was negligible. | Complete failure in establishing the design and the layout. |
| Preparation of report | A complete technical report perfect in all aspects such as appropriate literature review, clear work plan, valid experimental section, analysis and interpretation of the obtained results, plant design and layout, HAZOP studies, economic aspects and conclusion is presented. | A technical report with the literature review, work plan, valid experimental section, analysis and interpretation of the obtained results, plant design and layout, HAZOP studies, economic aspects and conclusion is presented. | An average technical report with literature review, work plan, a valid experimental section, plant design, and layout and conclusion are presented. | A report of the experimental works and analysis of the results is presented. | An incomplete report is presented. |
| Project presentation | A well-crafted presentation on the objectives, work plan, experimental design, results and discussion, plant layout and economic aspects done within the stipulated time limit. All the team members were fully confident to defend the queries raised by the panel. | A presentation on the objectives, work plan, experimental design, results and discussion, plant layout and economic aspects is done within the stipulated time limit. Team members defended the queries raised by the panel. | An average presentation on the objectives, work plan, experimental design, results and discussion, plant layout and economic aspects done within the stipulated time limit. Team members partially answered the queries raised by the panel. | An average presentation on the objectives, work plan, experimental design, results and discussion, plant layout and economic aspects done without proper time management. Team members could not answer the queries raised by the panel. | Presentation and its defense wasnot satisfactory. |

Once the project report is submitted it will be evaluated by the external panel constituted by the stakeholders (experts from industry and academia). The overall assessment of the report is done according to the specified criteria given in Table 8. Here also the grades were distributed as excellent (5), above average (4), average (3), below average (2), and poor (1). One of the critical components

in a project report is the effective communication through written and graphical representations of the data and analysis. If the results and analysis are not effectively communicated, the report will not be useful for the student's careers. Therefore, effective measures such as training, workshops, and seminars are provided for the students to succeed in articulating effective communication throughout the curriculum. Separate assessment criteria which are divided into a few factors as shown in Table 9 are given to external experts to measure the communication skills. The grading is done on a scale of 1–5.

**Table 8.** Evaluation criteria and grades.

| Criteria | Grade (1 to 5) |
| --- | --- |
| Motivation, Problem Statement, Objectives | |
| Literature survey | |
| Justification of the selected process | |
| Application of appropriate methodologies | |
| Selection and application of knowledge in science, mathematics and engineering principles | |
| Utilization of appropriate resources, tools, and software | |
| Graphical/chart presentation | |
| Data collection, analysis, interpretation, calculations, and conclusions | |
| Aspects of teamwork | |
| Future prospects | |

**Table 9.** Skills evaluation criteria in the report and grades.

| Assessment of Effective Communication through Written and Graphical Presentation | Grade (1 to 5) |
| --- | --- |
| Professional presentation of the report | |
| Organization of the report contents | |
| English formatting according to guidelines | |
| Effective use of pictures, models, figures, charts, tables, and graphs | |
| Proper references and citations in the report | |

Another important evaluation criterion is based on the oral presentation of the group. The external examiners formed to evaluate the report will judge the oral presentation too. The group will be provided with a stipulated time to present the project through the main points such as motivation for the work, identification of the problem, design of the project, experimental procedures and results, their analysis and logical conclusions, any future recommendations, and challenges and hurdles faced during the entire period of study. The criteria are divided into subgroups and each one is evaluated according to the grade scale from 1–5 which is given in Table 10. At the end of the presentation, the students will answer technical and non-technical questions related to the project. Furthermore, queries related to teamwork responsibilities and ethics are generally asked. The students will be individually evaluated with respect to their presentation and handling of the questions.

A short poster presentation on the project is also done at the end of the oral presentation, mostly in a public place. The group's performance is evaluated by the expert panel as well as the public audience, and the group will demonstrate the prototype or design in a professional way. Here also the group is evaluated individually. Both the oral presentation and the poster presentation carry 10 points towards calculating the final grades.

Another important aspect of the group project is the peer review, even though many experts believe that it cannot validate the actual results. The selected project is of almost 4 months duration, and by this time the participants will have adequate affiliations between them. Therefore, the grades given among them will not change much, and the minimal amount of points is given for peer review in this project (5 out of 100). The criteria for the peer review are given in Table 11.

**Table 10.** Oral Presentation Assessment by Evaluation Panel.

| | Grade (1 to 5) |
|---|---|
| **Organization** | |
| Well organized presentation highlighting the important tasks and findings | |
| **Presentation Delivery** | |
| Equal participation and equal responsibility in delivery of the presentation | |
| Fluency and confidence in presentation | |
| Presenter's knowledge in the selected technical area and ability to defend the questions | |
| Time management | |
| **Content** | |
| The project scope and problem statement | |
| Explanation of motivation | |
| Literature review and references | |
| Proper research methodology | |
| Experiments, data collection, analysis, interpretation and discussion | |
| Graphical/pictorial representation of the findings | |
| Conclusion and summary | |
| Hurdles faced during the execution of the project and the strategy adopted to overcome them | |
| **Questions and answer session** | |
| Ability to handle technical and non-technical queries | |
| Responsibilities of team members and team work related questions | |
| **Project poster presentation** | |
| Use of appropriate poster template | |
| Visual aids, legibility, & clarity | |
| Effectiveness in answering questions | |

**Table 11.** Allotment of scores by all members (students) of the team.

| Assessment Criteria | Grade (1 to 5) |
|---|---|
| Contribution: Do they attend, participate, share ideas, communicate and listen. | |
| Commitment: To common goal, do they keep on task and show concern for doing things properly, do they contribute to the team effort. | |
| Skill: Input, do they have added value or unique skills, do they show an understanding of ideas and apply them. | |
| Dependability: Reliable completion of tasks, do they show responsibility To the group and the tasks they need to do. | |
| Preparation: Do they come prepared and ready to work. | |
| Leadership: Do they help coordinate the overall project, find consensus, compromise. | |

Table 12 shows the average final grades of the evaluation of the designated project for a group of students in their final year in chemical engineering. The total points scored out of 100 were 79, which falls under the above average category. A careful examination of the grades given to the students reveals that the supervisor was needed to finalize the project tasks, including helping in the literature review, setting up the experimental design and its implementation, analysis of the obtained results,

and preparation of reports. Many of the current undergraduate students are not familiar with the methods required of the literature survey and compilation of the collected data even, though facilities are provided to them. Hence the supervisor has to encourage them to do the task. Also, the design of the experimental procedures and their implementation requires a certain level of supervision, as they lack experience in handling such projects. The analysis of the results and plant layout were done by the students, and fine-tuning of the results and layout was done by the supervisor.

**Table 12.** Final grades.

| Sl. No. | Task | Weightage (%) | Group Score |
|---------|------|---------------|-------------|
| 1. | Identifying the project task and defining the underlying problem | 10 | 8 |
| 2. | Literature review | 10 | 8 |
| 3. | Experimental design and implementation | 20 | 16 |
| 4. | Analysis of the results | 20 | 16 |
| 5. | Plant layout | 10 | 7 |
| 6. | Preparation of reports | 15 | 12 |
| 7. | Peer review | 5 | 5 |
| 8. | Project presentation | 10 | 7 |
| | Overall | 100 | 79 |

The report preparation and presentation depend on the contributions from each of the group members. If one of them fails to do successfully, the average value will go down. As can be seen from Table 11, the project report received a grade of above average, and the presentation received average grades from the examiners. Even though the report was prepared by the group together, the examiners found a few mistakes in formatting and spelling which lowered the grade to become above average. In the presentation, some of the students fared badly due to lack of proper practice. The peer review got a grade of excellent (5) which cannot be ruled out as the group has a better understanding among themselves and tries to help each other.

Overall, the current project obtained 79 out of 100 grade points, which can be considered as an above average performance as per the criteria set initially. The project assigned to the group of students contained many aspects of chemical engineering curricula and many of the theoretical aspects were experimentally tested and analyzed. They got a chance to analyze the results and implement and transform their knowledge into effective understanding and learning. In short, the presented project was a better choice for the students to help them learn the processes at the laboratory scale and the methods used to scale up the results considering all the necessary aspects of chemical engineering education. Such projects help the graduating students to achieve many of the student outcomes stipulated by accreditation agencies such as ABET in their criteria for accreditation of engineering programs [30] (Appendix B).

Similar studies have been reported in the literature in order to ascertain the improvement in skills needed for a graduating engineering student. Frank et al. reported the perceptions and attitudes of the mechanical engineering students after executing the project-based learning technique used in a certain course [31]. A detailed literature review on the PBL approach used on various engineering programs is given in the introduction of the report [32,33]. Similar to the study presented here, a small group of students was directed to conduct mini-projects which required design and experimentation. The authors concluded that the students acquired both interdisciplinary and multidisciplinary knowledge, many facets of engineering skills, and communication skills at both personal and interpersonal levels.

Uziak demonstrated the capabilities of the project-based learning approach in the engineering curriculum recently [34]. A fresh graduate from a normal educational institution often lacks the required knowledge and skill sets required for an industrial environment. The author explained the relevance of project-based learning to acquire the engineering skills and critical thinking ability needed to solve real-life problems. He has carefully reviewed the various reports based on project-based learning which enabled the students to acquire new knowledge apart from the curriculum through research activities related to the project, time management skills, and technical communication

capabilities [35]. The development of communication skills in various aspects like student to teacher, supervisor to the group, and students to student communication to achieve the best results for a particular problem is one of the major results of a project-based learning approach. This has also been established by the present project.

Another important aspect of the results of the current study is the development of lifelong learning methods for engineering students. When working in a group, members must communicate effectively, think logically and critically, understand the current status of the research problem, find out new and effective solutions, and improve individual skill sets according to the technological advances in their respective fields. Even though recycling is an old idea, effective solutions are still not available. The students are thus supposed to learn about current technologies, and even after graduation, they will have to continue to update their knowledge and skills to tackle this problem. The project does not end after the evaluation level; it extends throughout the student's life. Similar reports on the use of project-based learning for facilitating lifelong learning are available in the literature. For example, Lenschow reported on a large-scale project involving the industry, university, and colleges in a global setting to understand the lifelong learning processes [36]. The attitude of the students, knowledge of the supervisors and learning environment are critical to the promotion of lifelong learning capabilities for an engineering graduate. In this aspect, the present project achieved its goal, and the students were well-equipped to handle real-life problems in a smooth manner [37].

## 6. Conclusions

Project-based learning is one of the tools used to understand the learning capabilities of students in various types of educational streams. Conventional educational methods are not useful in the current technologically advanced world, and lately, many active learning methodologies are being proposed to ease the pressures of teaching and learning. A project-based learning approach was applied for the undergraduate students of a chemical engineering education program. It was found that students applied their acquired skills and knowledge to set up experimental designs and methodologies, analyzed the obtained results, proposed the layout for a commercial plant, and prepared the report according to the criteria put forward. The evaluation of the project was carried out at various stages by the panel (supervisor, external experts, and peers), and a total of 79 out of 100 points were given to the team. The project provided an excellent learning opportunity for the students by facilitating a real-time chemical engineering project with societal relevance.

The PBL approach can be applied in effective learning of process operations, design concepts, and laboratory activities of chemical engineering education programs. Apart from the attainment of the technical knowledge and skills, the PBL approach in the curriculum enables a graduating student to develop non-technical skills such as project management capabilities, communication skills, and presentation skills. By achieving technical and non-technical skills, a fresh engineer is fully equipped to handle real-world challenges.

**Author Contributions:** K.P.R., A.G. and S.P.T. equally contributed to the design and implementation of the research, to the analysis of the results, and to the writing of the manuscript.

**Funding:** This research received no external funding.

**Acknowledgments:** Authors are grateful to the management of Royal Commission Yanbu Colleges and Institutes division for their support and encouragement to apply project-based learning in the curriculum.

**Conflicts of Interest:** The authors declare no conflict of interest.

## Appendix A

**Calculations:**
$Q1$ = 48 g/min.
To get 1 ton output in a day of continuous production, $Q2$ should be 694 g/min
$D1$ = 19.05 mm

Q2 = Q1$(D_2/D_1)^2$
694 = 48$(D_2/D_1)^2$
$(D_2/D_1)^2$ = 14.46
$D_2/D_1$ = 3.80
Then $D_2$ = 3.80 × 19.05 = 72.4 mm
**Channel depth:**
$h_1$ = 5.2 mm
$h_2 = h_1\sqrt{\left(\frac{D_2}{D_1}\right)} = 5.2 \times \sqrt{(3.8)}$ = 10.14 mm
**Channel width:**
W1 = 12.7 mm
W2 = W1$(D_2/D_1)$ = 12.7 × (3.8) = 48.26 mm
**Screw speed:**
N1 = 250 rpm
So, N2 = N1$\sqrt{\left(\frac{D_2^2}{D_1}\right)} = 250 \times \sqrt{(3.8)}$ = 487.3 rpm
**Shear rate in the screw channel:**
$\gamma$1 = (π × D × N)/h = (3.14 × 19.05 × 250)/5.2 = 2876 min$^{-1}$
$\gamma$2 = $\gamma$1$\sqrt{\left(\frac{D_2}{D_1}\right)} = 2876 \times \sqrt{(3.8)}$ = 5606 min$^{-1}$
**Circumferential speed:**
V1 = π × D × N = 3.14 × 19.05 × 250 = 14,954 mm/min
So, V2 = V1$\sqrt{\left(\frac{D_2}{D_1}\right)} = 14{,}954 \times \sqrt{(3.8)}$ = 29,150 mm/min
**Residence time (tres):**
t1 = (π × D2 × L)/Q = [3.14 × (1.905)2 × 62.9]/34.78 = 20.6 min
t2 = t1$\sqrt{\left(\frac{D_2}{D_1}\right)} = 20.6 \times \sqrt{(3.8)}$ = 40.16 min
**Screw power:**
Z1 = 2.4 kW
Z2 = Z1 (D2/D1)2.5 = 2.4 (3.8)2.5 = 67.56 kW

**Appendix B**

ABET, Criteria for Accrediting Engineering Technology Programs, 2018–2019, General Criterion 3: Student Outcomes

a.　　an ability to select and apply the knowledge, techniques, skills, and modern tools of the discipline to broadly-defined engineering technology activities;

b.　　an ability to select and apply a knowledge of mathematics, science, engineering, and technology to engineering technology problems that require the application of principles and applied procedures or methodologies;

c.　　an ability to conduct standard tests and measurements; to conduct, analyze, and interpret experiments; and to apply experimental results to improve processes;

d.　　an ability to design systems, components, or processes for broadly-defined engineering technology problems appropriate to program educational objectives;

e.　　an ability to function effectively as a member or leader on a technical team;

f.　　an ability to identify, analyze, and solve broadly-defined engineering technology problems;

g.　　an ability to apply written, oral, and graphical communication in both technical and non-technical environments; and an ability to identify and use appropriate technical literature;

h.　　an understanding of the need for and an ability to engage in self-directed continuing professional development;

i.  an understanding of and a commitment to address professional and ethical responsibilities including a respect for diversity;

j.  a knowledge of the impact of engineering technology solutions in a societal and global context; and

k.  a commitment to quality, timeliness, and continuous improvement.

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
