# Peer review of "A Project Based Learning (PBL) Approach Involving PET Recycling in Chemical Engineering Education"

_recycling, doi:10.3390/recycling4010010_

Round 1
Reviewer 1 Report
The authors present a work where is evaluated Project based learning (PBL) approach in chemical engineering education, in case of PET bottles recycling. The article deals with current themes, which are very much discussed today and thus very interesting for the readers. Project based learning approach in connection with PET bottles recycling has a great future both in teaching and understanding technical practices and in educating the generation of people who will think more ecologically.
The introduction agrees and reflects well with the objectives of the work; however, for the better background of this topic, more references in the introduction is expected (about 5 references more).
In line 51 – in technical articles should not use contractions – won’t = will not
In line 60, 61 - remove the end of the paragraph, the sentence continues.
The chapter “Materials and Methods” need to be revised and supplemented with details of the process for better understanding.
In line 132 – “PET water bottles (1 liter capacity) were collected” – specify it, why 1 litre bottles just. What colour the PET bottles have, etc.?
In line 134 – What cutting machine (mill) was used?
In line 136-144 – The data can be better in the table, for easier imagination.
In line 151 – 120°C – upper index.
In the chapter “Project Results” results are written in table 3, 4 and 5; however, the recommendation is to provide a practical solution if there is any commercial solution to the extruder or a special design is needed. Next, what was used a granulation head (die plate), etc.
In line 164 – is correct – Speed N (rpm). Extra space – Time ( min).
In line 169-171 - there is no space between words
In line 171 – do not divide the table into two sides
In the chapter “Discussion” the tables have a very bad design, the reading is very complicated and confused. It is recommended to exchange columns to rows (Table 6). In table 7-10 can be written just number grades, the text is very long and then the design of the table is bad. Or can be written just one column for evaluation grades, where grade will be written as a number.
In conclusion, is need to add continuity and prospects for the future, what is expected and where to develop project based learning (PBL) approach in chemical engineering education.
Author Response
Dear Editor of Recycling Journal,
First of all, we thank the reviewers for their careful reading and thoughtful comments during this review processes. We would also like to thank the editor and editorial office for their time for processing the paper.
We have taken the reviewers comments into serious consideration in preparing our revised manuscript, which has resulted in a paper that is much clearer and meaningful. The following summarizes our response to reviewer comments point by point:
Comments and Suggestions for Authors
The authors present a work where is evaluated Project based learning (PBL) approach in chemical engineering education, in case of PET bottles recycling. The article deals with current themes, which are very much discussed today and thus very interesting for the readers. Project based learning approach in connection with PET bottles recycling has a great future both in teaching and understanding technical practices and in educating the generation of people who will think more ecologically.
The introduction agrees and reflects well with the objectives of the work; however, for the better background of this topic, more references in the introduction is expected (about 5 references more).
Answer: As suggested by the reviewer more references on project based learning is added in the introduction section. We have added six more recent references to highlight the importance of the selection of the PBL approach and objectives of the present work.
In line 51 – in technical articles should not use contractions – won’t = will not
Answer: As suggested corrections were made.
In line 60, 61 - remove the end of the paragraph, the sentence continues.
Answer: As suggested corrections were made.
The chapter “Materials and Methods” need to be revised and supplemented with details of the process for better understanding.
Answer: Line 166-172 describes the extrusion process in detail. Moreover, details about the raw material, cutting machine and process were added in the revised manuscript.
In line 132 – “PET water bottles (1 liter capacity) were collected” – specify it, why 1 litre bottles just. What colour the PET bottles have, etc.?
Answer: As suggested corrections were made. The water bottles were from Aquafina Company and were colorless. Such a selection of bottles was to ensure the uniformity of characteristics of the raw materials for the experimentation. This point is added in lines 150-152.
In line 134 – What cutting machine (mill) was used?
Answer: As suggested corrections were made. The details of the machine are provided in line 154-155.
In line 136-144 – The data can be better in the table, for easier imagination.
Answer: As suggested the data is inserted in to a table (Table 1).
In line 151 – 120°C – upper index.
Answer: Sorry for the mistake. As suggested, corrections were made.
In the chapter “Project Results” results are written in table 3, 4 and 5; however, the recommendation is to provide a practical solution if there is any commercial solution to the extruder or a special design is needed. Next, what was used a granulation head (die plate), etc.
Answer: As suggested corrections were made (line 200-204).
In line 164 – is correct – Speed N (rpm). Extra space – Time ( min).
Answer: Sorry for the mistake. As suggested, corrections were made.
In line 169-171 - there is no space between words
Answer: Sorry for the mistake. As suggested, corrections were made.
In line 171 – do not divide the table into two sides
Answer: As suggested, corrections were made by modifying the table properties.
In the chapter “Discussion” the tables have a very bad design, the reading is very complicated and confused. It is recommended to exchange columns to rows (Table 6). In table 7-10 can be written just number grades, the text is very long and then the design of the table is bad. Or can be written just one column for evaluation grades, where grade will be written as a number.
Answer: Table 7 (Table 6 before) describes the criteria for each grade. The tasks and the criteria for each grade are different and hence divided in to various separate rows. Even though, the table is long and appears as complicated, exchanging columns to rows will not simplify the design of the table. As suggested to change the evaluation grades to number grades, we have modified Tables 8-11 (Table 7-10 before) with one column for evaluation grades. The grades are given in scale of 1-5 in which 1 stands for poor and 5 stands for excellent as mentioned in the manuscript.
In conclusion, is need to add continuity and prospects for the future, what is expected and where to develop project based learning (PBL) approach in chemical engineering education.
Answer: Many thanks for such a valuable thought with respect to PBL approach and its future prospects. Accordingly the conclusion is enriched with few more sentences.

Reviewer 2 Report
Dear Authors, please improve your article with more research details and obtained results, compared with other similar experiments.
Author Response
Thank you very much for careful review of our manuscript. As suggested the manuscript has been revised with latest and similar works. We have enriched the manuscript with additional references and data.
Round 2
Reviewer 1 Report
The article was improved according to reviewer comments. It can be published in the present form.
I have no further comments on this article.
Author Response
Thank you very much for the review and comments. It really helped us to revise the manuscript and the quality of the manuscript improved a lot. We are glad that the reviewer is satisfied with our efforts. Once again thanks for the comments and second round of review.
Reviewer 2 Report
Dear Authors, please improve in details methodology and discus yours results in details with comparison to recent studies on related issues.
Author Response
The authors would like to thank the reviewer for the suggested modifications to the manuscript. The detailed methodology adopted in the present study is elaborated as section 3.1 in the revised manuscript. Moreover, a new figure (Figure 2) depicting the steps in the methodology is incorporated into the manuscript.
As suggested, an elaborated comparison of the results with recent relevant reports is included in the discussion section.
All the changes are marked in red in the revised manuscript.